# Community-based interventions targeting multiple forms of malnutrition among adolescents in low-income and middle-income countries: protocol for a scoping review

Adi Lukas Kurniawan ![ORCID],[1] Marijana Ranisavljev,[2] Uttara Partap ![ORCID] ,[3] Sachin Shinde,[3,4] Elisabetta Ferrero,[3] Sergej Ostojic,[5,6] Ntombizodumo Mkwanazi,[7] Deda Ogum Alangea,[8] Christine Neumann,[1] Shuyan Liu,[9] Till Bärnighausen,[1,10] Wafaie W. Fawzi ![ORCID] ,[3,11,12] ARISE-NUTRINT collaborators

**Correspondence to**
Dr Sachin Shinde;
sshinde@hsph.harvard.edu and
Dr Adi Lukas Kurniawan;
lukas.kurniawan@uni-heidelberg.de

## ABSTRACT

**Background** Adolescent malnutrition is a significant public health challenge in low-income and middle-income countries (LMICs), with long-term consequences for health and development. Community-based interventions have the potential to address multiple forms of malnutrition and improve the health outcomes of adolescents. However, there is a limited understanding of the content, implementation and effectiveness of these interventions. This scoping review aims to synthesise evidence on community-based interventions targeting multiple forms of malnutrition among adolescents in LMICs and describe their effects on nutrition and health.

**Methods and analysis** A comprehensive search strategy will be implemented in multiple databases including MEDLINE (through PubMed), Embase, CENTRAL (through Cochrane Library) and grey literature, covering the period from 1 January 2000 to 14 July 2023. We will follow the Participants, Concept and Context model to design the search strategy. The inclusion criteria encompass randomised controlled trials and quasi-experimental studies focusing on adolescents aged 10–19 years. Various types of interventions, such as micronutrient supplementation, nutrition education, feeding interventions, physical activity and community environment interventions, will be considered. Two reviewers will perform data extraction independently, and, where relevant, risk of bias assessment will be conducted using standard Cochrane risk-of-bias tools. We will follow the PRISMA Extension for Scoping Reviews checklist while reporting results.

**Ethics and dissemination** The scope of this scoping review is restricted to publicly accessible databases that do not require prior ethical approval for access. The findings of this review will be shared through publications in peer-reviewed journals, and presentations at international and regional conferences and stakeholder meetings in LMICs.

**Scoping review registration** The final protocol was registered prospectively with the Open Science Framework on 19 July 2023 (https://osf.io/t2d78).

## STRENGTHS AND LIMITATIONS OF THIS STUDY

⇒ A comprehensive examination of over 20 years of published data will be conducted.
⇒ Grey literature sources such as government reports and organisation websites will also be included.
⇒ There will be a quality assessment of the included quantitative studies.
⇒ The proposed search strategy will be conducted only in three electronic databases.

## INTRODUCTION

The current global adolescent population surpasses 1.2 billion individuals, with approximately 90% of them residing in low-income and middle-income countries (LMICs).[1] Moreover, when considering regional trends, it is expected that the proportion of young people aged 10 to 24 residing in Sub-Saharan Africa will experience substantial growth, rising from 245 million in 2015 to 605 million by 2050.[2] In contrast, the Asia and Pacific region is expected to undergo the most significant decrease, declining from 718 million in 2015 to 619 million by 2050.[2] These regional variations emphasise the unique challenges and opportunities faced by different regions in terms of future demographic and health challenges, which will require distinct solutions.

Adolescence is a period of rapid physical growth, cognitive development, socioemotional development and cultural development, all of which are strongly influenced by an individual's socioeconomic, cultural and physical environments.[3] Nutrition plays a crucial role in improving health and development during this critical stage

in life, bringing intergenerational benefits. After the first 1000 days of life, adolescence is assumed to offer a second window of opportunity for correcting nutritional deficiencies and insufficient growth since childhood.[4]

LMICs are experiencing a rapid nutrition transition among adolescents, accompanied on the one hand by stunting, thinness, anaemia and other micronutrient deficiencies, and on the other hand, by an increasing burden of obesity and non-communicable diseases.[5] Malnutrition was the leading cause of disability-adjusted life years among the 10–14 age group in 2019, followed by iron deficiency anaemia among adolescents aged 10–19.[6] The consumption of diverse and healthy diets by adolescents from LMICs is declining, while the consumption of processed and calorie-rich foods is on the rise, contributing to rising obesity rates.[7] Furthermore, food insecurity has been aggravated in vulnerable populations including adolescents in LMICs because of the COVID-19 pandemic, political instability, and recurring climate crises in the form of flooding and droughts.[8]

Several systematic reviews indicate that micronutrient supplementation is effective in addressing nutritional deficiencies.[9 10] Iron supplementation can reduce anaemia in adolescents; periconceptional folic acid supplementation among adolescent girls can reduce neural tube defects; and adolescent girls who consume high amounts of calcium (≥1 g daily) have lower rates of preeclampsia, preterm birth or neonatal hospitalisation.[11] There is limited evidence that protein-energy supplements are effective for adolescents.[10]

The burden of malnutrition may be reduced by several nutrition-sensitive interventions, including nutrition education, dietary interventions, physical activity and food environment interventions.[12] Several systematic reviews suggest promising but modest results from discrete nutrition-sensitive interventions aimed at addressing malnutrition in schools.[13–15] These single-domain interventions, however, target either undernutrition or overnutrition and operate in *silos*. There is increased interest in addressing health and nutrition behaviours through integrated interventions, generally called 'double-duty actions', targeting multiple forms of malnutrition and nutrition-related non-communicable diseases.[16] An essential element of this concept is that tackling one form of malnutrition should not prevent addressing another. There is promising evidence that integrated interventions can improve the nutritional status of school-going children and adolescents.[12]

Nevertheless, there are several gaps in understanding adolescent nutrition in LMICs. Currently, nutrition-specific and nutrition-sensitive interventions tend to focus on school-going adolescents, and little is known about their effects on other vulnerable groups of adolescents, such as out-of-school adolescents, migrant adolescents and HIV-positive adolescents. Another important gap is that most of the school-based interventions target overlapping age groups and there is little known regarding the age-appropriate intervention strategies and delivery

mechanisms as well as the specific impact on the adolescent population. Moreover, most of these school-based interventions are delivered by schoolteachers, community health workers, school nurses or peers in classroom-based settings or during school hours. Despite the importance of nutrition-sensitive and nutrition-specific interventions for the health of communities, little evidence exists about their form and function using community platforms.

The purpose of this scoping review is to comprehensively review the literature to describe community-based interventions that address the multiple forms of malnutrition such as obesity, overweight, underweight, wasting, stunting, anaemia and micronutrient deficiencies affecting adolescents in LMICs and describe the effects of these interventions on nutrition and health. We decided to conduct a scoping review as our primary aim was to summarise the overview of the evidence on community-based interventions for adolescents in LMICs rather than to pursue a specific clinical or epidemiological question related to these or provide evidence to directly inform policy or practice.[17] In the context of this review, community-based interventions refer to any interventions carried out in community settings other than schools to improve the health among adolescents. Examples include interventions implemented through community youth centres, clubs or religious centres. By excluding school-community interventions, which have been thoroughly explored in the literature, we can concentrate on interventions that are less common, less understood and less easy to implement but that have the potential to reach the most vulnerable groups of adolescents.

## METHODS
### Data sources, search terms, and search strategy

As part of our primary strategy, we will search MEDLINE (through PubMed), Embase and CENTRAL (through the Cochrane Library). All databases will be searched for eligible studies from 1 January 2000 through 14 July 2023. We will identify potentially relevant published studies using the combination of medical subject headings (MeSH) and text words denoting nutrition-specific and nutrition-sensitive interventions. We will also examine references and bibliographies of included studies to identify additional sources of information. This search of studies will be supplemented by reviewing ClinicalTrials. gov and organisational websites such as the WHO, World Bank, UNICEF and United Nations Population Fund. When possible, reports written in languages other than English will be translated by colleagues who are native speakers of those languages. No study will be considered if it cannot be adequately translated.

We will use the Participants, Concept and Context model (table 1) to guide our search strategy. The search will use indexing terms, including MeSH terms, keywords and free text words. First, a broad search strategy (eg, type of study (randomised controlled trials, quasi-experiments or controlled before-after studies) AND intervention domain

**Table 1** Eligibility criteria for the scoping review

| Item | Inclusion criteria | Exclusion criteria |
|---|---|---|
| Participants | Studies involving adolescents (10–19 years old) | Studies involving children <10 years of age or adults (>19 years of age) |
| Concept | Studies involving one or more of the following interventions: nutrient supplementation interventions including vitamin and nutrient supplementation, deworming, complementary feeding, nutrition education, physical education, promoting healthy diets and/or physical activity, nutrition policies, community/home garden, water, sanitation and hygiene interventions, community environment interventions, and structural interventions such as sweetened beverage tax, soda tax and sugary drink tax. Studies that compared the intervention with any relevant control group including comparisons with no intervention, regular nutrition education and/or physical education, or any other intervention in the community setting | Interventions targeted towards individuals with specific medical conditions such as treatments intended for underweight, overweight or obese adolescents |
| Context | Community settings in low-income and middle-income countries | Interventions applied exclusively in the school setting |
| Types of sources | Randomised controlled trials, quasi-experimental studies including controlled before-after studies | Non-randomised trials including controlled before-after studies that did not account for baseline differences, observational studies including cohort, case-control and cross-sectional designs, and editorial commentaries, opinions and review articles |

(eg, nutrition education) AND population (adolescents) AND setting (low-income and middle-income countries)) will be performed in PubMed. We will confirm the sensitivity of the search strategy by identifying several sentinel articles. The PubMed strategy, provided in online supplemental file 1, will be adapted to suit other databases. We will document the following details for each search: databases searched, date of search, search strategy (ie, subject headings and keywords, including if terms are expanded, truncated and how they are combined), filters used and the number of records retrieved. Additionally, a source will be provided for each publication identified through manual search (ie, journal name, website, conference proceedings, etc).

### Eligibility
The inclusion and exclusion criteria for this scoping review are listed below.

### Inclusion criteria
We will include the following studies.
► Randomised controlled trials (RCT), with the intervention randomised to individuals or in clusters (including clubs, groups, communities, villages, homes, etc), and quasi-experimental studies including controlled before-after studies that have reported interventions to address any form of adolescent malnutrition when compared with a control group.
► Studies involving adolescent boys and/or girls aged 10–19 years, based on the WHO definition of adolescents.[18]

► Studies conducted in LMICs—as defined by the World Bank in the year 2023.[19]
► Studies involving interventions for one or more of the following: micronutrient supplementation, feeding interventions, nutrition education, physical education, interventions to promote healthy diets, interventions promoting physical activity, community and/or home gardens, food and nutrition policies, community environment interventions, water sanitation and hygiene interventions, and structural interventions such as taxation of sweetened-sugary drinks.
► The control (comparison) in each included study can be participants who did not receive any intervention or received standard care, received standard health/nutrition education or any other intervention in the community setting.
► Published articles as well as unpublished and grey literature and will include ongoing studies where preliminary findings are available.
► We will not place any restrictions on the language, sample size or duration of the intervention.

### Exclusion criteria
We will not consider the following studies.
► Non-RCTs that are not quasi-experimental studies with comparator groups and controlled before-after studies that did not account for the baseline differences between the study arms.
► Observational studies such as cohort, case-control, and cross-sectional designs.

- ► Editorials, commentaries, opinions and review articles. However, we will use review articles to identify additional original articles.
- ► Studies that were conducted in the school setting and clinical interventions targeted individuals with specific medical conditions such as programmes intended for underweight, overweight, obese or anaemic adolescents.

## Data management

The records will be imported into Covidence (Veritas Health Innovation, Melbourne, Australia), an Internet-based systematic review management programme. Detection and removal of duplicates, title and abstract screening, and full-text screening will be performed by using Covidence.

## Selection of studies

Using Covidence, we will screen titles, abstracts and full texts. First, two reviewers will independently assess all search results (ie, titles and abstracts) and exclude irrelevant studies based on inclusion and exclusion criteria. Next, two reviewers will carry out the full-text screening based on the same inclusion/exclusion criteria. The reviewers will discuss and resolve any difference of opinion or, if necessary, seek a third reviewer's opinion for resolving differences. A study flow diagram stating the specific reasons for exclusion will be maintained following the PRISMA for Scoping Review statement (PRISMA-ScR).[20]

## Data extraction

Two reviewers will independently extract and enter data from studies included in the review. We will develop and test an extraction form on five randomly selected studies. We will extract the following information.

- ► Study details including the title, authors (first author and corresponding author), the corresponding author's contact information, journal (or source for unpublished reports), calendar year of publication, calendar year of intervention, country and source of funding.
- ► Study methods including objectives and/or research questions, type of study, investigation strategies, settings, sample size and sample characteristics (eg, age, sex, socioeconomic status
- ► Intervention strategy including target population, delivery platform and providers (including selection, training, supervision, support and incentivisation), types of nutrition and other interventions (including content, conceptual framework and/or theoretical underpinnings, timing, duration, and dosage or frequency) and comparator/control.
- ► Outcomes assessed and details of the measures used.
- ► Findings including the coverage of services, facilitators and barriers to intervention delivery and uptake, effectiveness findings with point estimates and measures of variance (standard errors, 95% CI or p values),

and any other key findings related to the scoping review questions.

We will contact the corresponding author via email if there is missing or inconsistent information. We will contact the author two times at most. The available data will be analysed and any gaps due to missing data will be discussed if the data issue cannot be resolved after contacting the authors. The extraction form template was provided in online supplemental file 2.

## Risk of bias assessment

As scoping reviews are exploratory in nature, risk of bias assessments are not typically required as part of the guidelines for scoping reviews.[17] However, we plan to assess the risk of bias among studies with an available quantitative measure as a preliminary way of contextualising the reported measures of impact on the outcomes reported. For the assessment of the risk of bias in the selected studies, we will use the Cochrane Collaboration's revised tool for assessing the risk of bias in randomised trials (RoB 2).[21] Two reviewers will independently evaluate methodological quality. Any uncertainties or disagreements will be resolved by discussion or by a third reviewer, whenever needed. The tool is a domain-based evaluation, in which critical assessments for risk of bias are made separately for various domains, including the randomisation process, deviation from intended interventions, missing outcome data, measurement of the outcome and selective outcome reporting. The risk of bias in clustered trials will be similarly assessed using the risk of bias 2 for cluster-randomised trials (RoB 2 CRT).[22] Additionally, we will use the Risk of Bias in Non-randomised Studies of Interventions (ROBINS-I) tool[23] to assess the risk of bias for controlled before-after studies and non-randomised controlled trials.

## Synthesis of evidence

All included studies will be systematically synthesised in the text and a table following the SWiM guidelines (Synthesis Without Meta-analysis).[24] In this synthesis, we will describe how many sources of evidence were screened, assessed for eligibility and included in the review, along with reasons for exclusion at each stage. Our presentation of included sources of evidence will include summary characteristics and citations, as well as a critical appraisal, if applicable. Studies will be grouped based on methods and interventions, standardised outcome metrics, synthesis methods, criteria used to prioritise results for summary, reporting of results, the certainty of results, heterogeneity in effects, as well as barriers and facilitators to delivering the interventions will be discussed. For continuous outcomes, effect estimates will be expressed as mean differences (with 95% CI) between the intervention group and the control group; for dichotomous outcomes, effect estimates will be expressed as risk ratios, rate ratios, hazard ratios or ORs (all with a 95% CI). Additionally, we will discuss the limitations of the review process and provide an interpretation of the results concerning the objectives

of the review, as well as possible implications or next steps. We will follow the PRISMA Extension for Scoping Reviews (PRISMA-ScR) checklist and guidelines to ensure a robust and replicable process.[20]

### Registration and reporting

The final protocol was registered prospectively with the Open Science Framework (https://osf.io/t2d78) on 19 July 2023, based on the PRISMA-ScR.[20] In the event of protocol amendments, the date of each amendment will be accompanied by a description of each change and the rationale on the Open Science Forum.

### Ethics and dissemination

This study is a scoping review that does not require ethics approval because it involves a methodical presentation of available resources. The protocol aims to provide an overview of the broad literature on community-based interventions targeting multiple forms of malnutrition among adolescents in LMICs. We anticipate that the findings of this review will be disseminated through publications in peer-reviewed journals, and presentations at international and regional conferences and stakeholder meetings targeting researchers, adolescents, policymakers and governments in LMICs. Additionally, the submitted review will help identify effective interventions, determine gaps and disparities among interventions, and provide insight for policymakers to develop and design as well as implement future programmes.

### Patient and public involvement

None. This work analyses existing research studies, and therefore, involves no patients or members of the public.

**Author affiliations**
[1]Heidelberg Institute of Global Health, Heidelberg University Hospital, Heidelberg, Germany
[2]Faculty of Sport and Physical Education, University of Novi Sad, Novi Sad, Serbia
[3]Department of Global Health and Population, Harvard T. H. Chan School of Public Health, Boston, Massachusetts, USA
[4]Center for Inquiry into Mental Health, Pune, Maharashtra, India
[5]Faculty of Medicine, University of Novi Sad, Novi Sad, Serbia
[6]Department of Nutrition and Public Health, University of Agder, Kristiansand, Norway
[7]Research Division, University of KwaZulu-Natal, Durban, South Africa
[8]Department of Population, Family, and Reproductive Health, School of Public Health, University of Ghana, Accra, Ghana
[9]Department of Psychiatry and Psychotherapy, Charité - Universitätsmedizin Berlin, Berlin, Germany
[10]Africa Health Research Institute, Durban, South Africa
[11]Department of Nutrition, Harvard T. H. Chan School of Public Health, Boston, Massachusetts, USA
[12]Department of Epidemiology, Harvard T. H. Chan School of Public Health, Boston, Massachusetts, USA

**Collaborators** ARISE-NUTRINT collaborators: Michael Laxy, Professorship of Public Health and Prevention, Technical University of Munich, Germany. Jacob Burns, Professorship of Public Health and Prevention, Technical University of Munich, Germany. Sara Pedron, Professorship of Public Health and Prevention, Technical University of Munich, Germany. Mary Mwanyika Sando, Africa Academy for Public Health, Tanzania. Ayoade Oduola, University of Ibadan Research Foundation, Nigeria. Mosa Moshabela, University of KwaZulu-Natal, South Africa. Ali Sié,

National Institute of Public Health, Nouna Health Research Center, Burkina Faso. David Guwatudde, School of Public Health, Makerere University, Uganda. Yemane Berhane, Addis Continental Institute of Public Health, Ethiopia. Adom Manu, Department of Population, Family, and Reproductive Health, University of Ghana, Ghana. Jan A.C. Hontelez, Erasmus Universitair Medisch Centrum, The Netherlands. Magda Rosenmöller, Center for Research in Healthcare Innovation Management, IESE Business School, Spain. Irene Brandt, Heidelberg Institute of Global Health, Heidelberg University Hospital, Germany. Ina Danquah, Heidelberg Institute of Global Health, Heidelberg University Hospital, Germany. Matthias Kern, Heidelberg Institute of Global Health, Heidelberg University Hospital, Germany. Joy Mauti, Heidelberg Institute of Global Health, Heidelberg University Hospital, Germany. Shannon McMahon, Heidelberg Institute of Global Health, Heidelberg University Hospital, Germany. Japhet Killewo, Muhimbili University of Health and Allied Sciences, Tanzania. Hanna Y. Berhane, Department of Nutrition and Behavioral Sciences, Addis Continental Institute of Public Health, Ethiopia. Amani Tinkasimile, Africa Academy for Public Health, Tanzania. Mashavu Yussuf, Africa Academy for Public Health, Tanzania. Innocent Yusufu, Africa Academy for Public Health, Tanzania. Laetitia Paumard, Center for Research in Healthcare Innovation Management, IESE Business School, Spain. Millogo Ourohiré, National Institute of Public Health, Nouna Health Research Center, Burkina Faso. Erick Agure, Heidelberg Institute of Global Health, Heidelberg University Hospital, Germany. Jabulani Ncayiyana, College of Health Sciences, University of KwaZulu-Natal, South Africa. Bruno Sunguya, School of Public Health and Social Sciences, Muhimbili University of Health and Allied Sciences, Tanzania. Tiwatayo Lasebikan, Center for Research in Healthcare Innovation Management, IESE Business School, Spain. Marina Taonda, National Institute of Public Health, Nouna Health Research Center, Burkina Faso. Sylvain Somé, National Institute of Public Health, Nouna Health Research Center, Burkina Faso. Clarisse Dah, National Institute of Public Health, Nouna Health Research Center, Burkina Faso. Katian Napon, National Institute of Public Health, Nouna Health Research Center, Burkina Faso. Moussa Ouédraogo, National Institute of Public Health, Nouna Health Research Center, Burkina Faso.

**Contributors** SS conceived the idea, developed the methods and wrote the first draft of the manuscript. ALK, MR, UP and EF contributed to the methods and supported the drafting and editing of the manuscript. ALK and MR contributed meaningfully to the design of the search strategy. SO, NM, DOA, CN, SL, TB, WWF and ARISE-NUTRINT study collaborators supervised and reviewed the protocol. All authors revised and approved the final manuscript.

**Funding** This study was funded by the European Union Horizon 2022. Views and opinions expressed are however those of the author(s) only and do not necessarily reflect those of the European Union. Neither the European Union nor the granting authority can be held responsible for them.

**Competing interests** None declared.

**Patient and public involvement** Patients and/or the public were not involved in the design, or conduct, or reporting, or dissemination plans of this research.

**Patient consent for publication** Not applicable.

**Ethics approval** Not applicable.

**Provenance and peer review** Not commissioned; externally peer reviewed.

**Data availability statement** Data sharing not applicable as no datasets generated and/or analysed for this study.

**ORCID iDs**
Adi Lukas Kurniawan http://orcid.org/0000-0002-9809-1953
Uttara Partap http://orcid.org/0000-0002-2531-1804

Wafaie W. Fawzi http://orcid.org/0000-0002-2908-600X

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
