## [Reviewer comments · BMJ Open]

ARTICLE DETAILS

TITLE (PROVISIONAL)	Community-based interventions targeting multiple forms of malnutrition among adolescents in low- and middle-income countries: protocol for a scoping review
AUTHORS	Kurniawan, Adi Lukas; Ranisavljev, Marijana; Partap, Uttara; Shinde, Sachin; Ferrero, Elisabetta; Ostojic, Sergej; Mkwanzai, Ntombizodumo; Alangea, Deda Ogum; Neumann, Christine; Liu, Shuyan; Bämighausen, Till; Fawzi, Wafaie.; Collaborators, ARISE-NUTRINT

VERSION 1 – REVIEW

REVIEWER	Khamis, Ahmed Gharib Muhimbili University of Health and Allied Sciences
REVIEW RETURNED	17-Sep-2023

GENERAL COMMENTS	GENERAL COMMENTS This paper is not well written and the general presentation of ideas and level of english is good. However, if the objective of this review is to assess community-based intervention, in LMICs authors should provide meaning and context of the word “community-based”. OTHER COMMENTS Abstract Weel elaborated, the information provided is enough Line 77: “There is lack”. I prefeere to say there is limited. Introduction The introduction section is well drafted Line 122: It is commonly known as “Second window of opportunity”. Please correct. Provide meaning and definition of community based interventions? Materials and Methods How grey literature will be searched and reported? Any strategy? What about SAGE database in your search? But school is part of the community is LMIC context, the reason of excluding school is not well clear. Any justification for choosing 10-19 range? Provide a reference Line 197 : “We will include”, delete the words Line 213: “We will include”, delete the words It will be appropriate to attach the sample of extraction form Discussion Line 309: “Multiple forms”, please correct Line 311-319: Justify that school environment is part of the community-based interventions. This is against your methodology
---

REVIEWER	Mayasari, Noor Rohmah State University of Surabaya, Nutrition
REVIEW RETURNED	21-Oct-2023

GENERAL COMMENTS	The protocols have significant benefits as consideration for intervention targeting multiple of malnutrition among adolescents in low- and middle-income countries. The authors need to clarify some comments below:  1. Why exclude school-based intervention. Is it crucial? 2. According to the title, the outcome is supposed to be multiple deficiencies. I don't think knowledge of diet and nutrition, dietary intake (i.e. amount and frequency), dietary diversity, diet quality, physical activity, sedentary behaviors, nutrition literacy, and nutrition fluency are appropriate to be included as outcomes. 3. Study design excludes cohort studies, Some studies such as Intervention tax may use prospective cohort studies as the design.
---

REVIEWER	Silva, Flávia UFCSPA, NUTRITION
REVIEW RETURNED	21-Jan-2024

GENERAL COMMENTS	This scoping review aims to synthesize evidence on community-based interventions targeting multiple forms of malnutrition among adolescents in LMICs and describe their effects on nutrition and health. Thank you for the opportunity to review this manuscript. See below my comments:  1. Abstract  - Please, include the PCC acronyms in the methods. - Cite that grey literature will also be accessed. - Adjust the abstract according to the comments related to the methods. - What do you mean, where it will be relevant, risk of bias will be accessed? 2. Strengths and limitations  - Why quality assessment will be performed only for studies with a impact measure? 3. Introduction  - Page 8, line 134: You say, "several systematic reviews" and cite only one reference that is not a systematic review. Please, adjust and include the original references. - Page 8, line 138: Include a reference for the following sentence "There is limited evidence that protein-energy supplements are effective for adolescents". 4. Methods  - Do you intend to conduct a scoping review or a systematic review? There are important differences among them, and your methods is not in accordance with a scoping review. Please, see the guidance of Joanna Briggs Institute for scoping reviews. Some points: PICO is not appropriate for scoping reviews; the recommended reporting guideline is PRISMA ScR instead of SWiN (as cited in the abstract and Methods). - How did you define the eligible interventions? Will be the study included if it combined more than one intervention? - Why will you use Endnote and Covidence for the same purpose? - Where the data extraction will be performed? - In the OSF is described as end date for this scoping review December 2023. Did you have already finished the study? - Please, include the date when the PubMed search was conducted, and the results obtained.
---

VERSION 1 – AUTHOR RESPONSE

Reviewer 1:

GENERAL COMMENTS

This paper is not well written and the general presentation of ideas and level of English is good. However, if the objective of this review is to assess community-based intervention, in LMICs authors should provide meaning and context of the word “community-based”.

Response: Thank you for your kind words and comments, which have further strengthened this manuscript. In the revised manuscript, we have included the context of and definition of community-based interventions.

Introduction, lines 166 – 172:

"In the context of this review, community-based interventions refer to any interventions carried out in community settings other than schools, to improve the health among adolescents. Examples include interventions implemented through community youth centers, clubs, or religious centers. By excluding school-community interventions, which have been thoroughly explored in the literature, we can concentrate on interventions that are less common, less understood, and less easy to implement, but that have the potential to reach the most vulnerable groups of adolescents."

OTHER COMMENTS

Abstract

Well elaborated, the information provided is enough

Line 77: “There is lack”. I prefer to say there is limited.

Response: This line is corrected as follows:

Abstract, lines 77 – 78:

"However, there is a limited understanding of the content, implementation, and effectiveness of these interventions."

Introduction

The introduction section is well-drafted

Line 122: It is commonly known as “Second window of opportunity”. Please correct.

Provide meaning and definition of community-based interventions.

Response: Our revision of line 122 reflects the suggestion. We also defined community-based interventions as follows.

Introduction, lines 118 – 120:

"After the first 1000 days of life, adolescence is assumed to offer a second window of opportunity for correcting nutritional deficiencies and insufficient growth since childhood (4).

Introduction, lines 166 – 172:

"In the context of this review, community-based interventions refer to any interventions carried out in community settings other than schools, to improve the health among adolescents. Examples include interventions implemented through community youth centers, clubs, or religious centers. By excluding school-community interventions, which have been thoroughly explored in the literature, we can concentrate on interventions that are less common, less understood, and less easy to implement, but that have the potential to reach the most vulnerable groups of adolescents."

Materials and Methods

How grey literature will be searched and reported? Any strategy?

Response: The primary search focus is on the published literature, using databases like MEDLINE, EMBASE, and Cochrane Library. Additionally, we will explore relevant websites where pertinent reports may be found to complement these main searches. Among them would be clinical trial registries as well as websites of UN agencies.

Methods, lines 176 - 183:

"As part of our primary strategy, we will search MEDLINE (through PubMed), Embase, and CENTRAL (through the Cochrane Library). [...] This search of studies will be supplemented by reviewing ClinicalTrials.gov and organizational websites such as the World Health Organization (WHO), World Bank, United Nations Children's Fund (UNICEF), and United Nations Population Fund (UNFPA)."

What about SAGE database in your search?

Response: Thank you for your suggestion. We have opted not to utilize the SAGE database, as it exclusively comprises journals published by the SAGE publishing company. This database is not widely employed in systematic or scoping reviews, as highlighted in a recent paper by Gusenbauer (2020), which assesses databases commonly utilized for reviews, both in published and grey literature. Additionally, all significant journals published by SAGE are indexed in the databases we have selected for our search.

Gusenbauer M, Haddaway NR. Which academic search systems are suitable for systematic reviews or meta-analyses? Evaluating retrieval qualities of Google Scholar, PubMed, and 26 other resources. Res Synth Methods. 2020;11:181-217 PubMed . doi: 10.1002/jrsm.1378.

But school is part of the community in LMIC context, the reason of excluding school is not well clear.

Response: Thank you for this suggestion. To address this point, we added a sentence to clarify why we excluded schools. This is because nutrition school-based interventions are very common and thoroughly examined in the literature (Shinde et al., 2023). In contrast, community-based interventions are less common and less easy to implement, but they have enormous potential because they can reach the most vulnerable groups of adolescents. Focusing on these types of interventions will help us

better understand how they work and how they can be effectively implemented, to promote a higher uptake in the future. The addition is as follows:

Introduction, lines 169 – 172:

"By excluding school-community interventions, which have been thoroughly explored in the literature, we can concentrate on interventions that are less common, less understood, and less easy to implement, but that have the potential to reach the most vulnerable groups of adolescents."

Shinde S, Wang D, Moulton GE, et al. School-based health and nutrition interventions addressing double burden of malnutrition and educational outcomes of adolescents in low- and middle-income countries: A systematic review. *Matern Child Nutr.* 2023 30:e13437. doi: 10.1111/mcn.13437.

Any justification for choosing 10-19 range? Provide a reference

Response: We chose the 10-19 years age range based on the WHO classification of adolescents (Singh, 2019). We added this information to the protocol.

Methods, lines 209 - 210: *"Studies involving adolescent boys and/or girls aged 10-19 years, based on the WHO definition of adolescents (Singh et al., 2019)."*

Singh JA, Siddiqi M, Parameshwar P, et al. World Health Organization Guidance on Ethical Considerations in Planning and Reviewing Research Studies on Sexual and Reproductive Health in Adolescents. *J Adolesc Health.* 2019;64:427-429 PubMed . doi: 10.1016/j.jadohealth.2019.01.008.

Line 197 : "We will include", delete the words

Line 213: "We will include", delete the words

Response: We deleted these words from the text in line 205 and line 220.

It will be appropriate to attach the sample of the extraction form

Response: We have included the sample of the extraction form as a supplementary file 2

Discussion

Line 309: "Multiple forms", please correct

Response: This is corrected in the revised manuscript. However, we have now removed the discussion section of our protocol, per the editor's suggestion.

Line 311-319: Justify that the school environment is part of the community-based interventions. This is against your methodology

Response: We have now removed the discussion section of our protocol, per the editor's suggestion, but with this paragraph, we wanted to emphasize that, while schools are important platforms for

delivering health and nutrition interventions for adolescents, there is a large proportion of out-of-school adolescents who are most vulnerable and can be reached through community-based interventions.

Reviewer 2:

The protocols have significant benefits as consideration for intervention targeting multiple of malnutrition among adolescents in low- and middle-income countries. The authors need to clarify some comments below:

1. Why exclude school-based intervention. Is it crucial?

Response: Thank you for this suggestion. It is important to exclude school-based interventions for the purposes of our review because nutrition school-based interventions are very common and thoroughly examined in the literature (Shinde et al., 2023). In contrast, community-based interventions are less common and less easy to implement, but they have enormous potential because they can reach the most vulnerable groups of adolescents. Focusing on these types of interventions will help us better understand how they work and how they can be effectively implemented, to promote a higher uptake in the future. We added a definition of community-based interventions for the purpose of our review and explained why we excluded school-based interventions towards the end of the introduction section. The addition is as follows:

Introduction, lines 166 – 172:

"In the context of this review, community-based interventions refer to any interventions carried out in community settings other than schools, to improve the health among adolescents. Examples include interventions implemented through community youth centers, clubs, or religious centers. By excluding school-community interventions, which have been thoroughly explored in the literature, we can concentrate on interventions that are less common, less understood, and less easy to implement, but that have the potential to reach the most vulnerable groups of adolescents."

Shinde S, Wang D, Moulton GE, et al. School-based health and nutrition interventions addressing double burden of malnutrition and educational outcomes of adolescents in low- and middle-income countries: A systematic review. *Matern Child Nutr.* 2023 30:e13437. doi: 10.1111/mcn.13437.

2. According to the title, the outcome is supposed to be multiple deficiencies. I don't think knowledge of diet and nutrition, dietary intake (i.e. amount and frequency), dietary diversity, diet quality, physical activity, sedentary behaviors, nutrition literacy, and nutrition fluency are appropriate to be included as outcomes.

Response: Thank you for this comment. The term "multiple forms of malnutrition" in our title encompasses a broad definition, including micronutrient deficiencies as well as manifestations of malnutrition such as thinness (low BMI), overweight and obesity (high BMI), and anemia. We agree that diet- and physical activity-related behaviours do not necessarily form part of the multiple forms of malnutrition – rather, they lie on the pathway to such outcomes. Regardless, we have not specified such outcomes of interest in the search, nor did we restrict our search by these outcomes, in order to ensure the most inclusive search possible. We have now edited Table 1 accordingly and revised it in order to take into account the PCC (participants-concept-context) framework that is more standard for scoping reviews and clarifies that there is no restriction by the outcome.

Please see the revised Table 1

3. Study design excludes cohort studies, Some studies such as Intervention tax may use prospective cohort studies as the design.

Response: The Reviewer makes an important point regarding whether and how specific interventions would be included. The specific example that the Reviewer has highlighted (and other similar examples where an intervention is being studied but has not been assigned in a randomized manner across groups) is classified as a quasi-experimental design (similar to other previously published studies (Jackson et al., 2023). It would therefore be eligible under our scoping review criteria and included as part of our review. Indeed, we are interested in studies that examine taxes as interventions, as noted in Table 1.

Jackson KE, Hamad R, Karasek D, et al. Sugar-Sweetened Beverage Taxes and Perinatal Health: A Quasi-Experimental Study. *Am J Prev Med.* 2023;65:366-376 PubMed . doi: 10.1016/j.amepre.2023.03.016.

Reviewer 3:

1. Abstract

- Please, include the PCC acronyms in the methods.
- Cite that grey literature will also be accessed.
- Adjust the abstract according to the comments related to the methods.
- What do you mean, where it will be relevant, risk of bias will be accessed?

Response: Thank you for this suggestion. We have adjusted the abstract accordingly. Moreover, when referring to the assessment of risk of bias, this refers specifically to the assessment process used exclusively in the context of quantitative studies.

Abstract, lines 82 – 84:

“...and grey literature, covering the period from January 1, 2000, to July 14, 2023. We will follow the Participants, Concept, and Context (PCC) model to design the search strategy.”

2. Strengths and limitations

Why quality assessment will be performed only for studies with a impact measure?

Response: Thank you for this comment. As this is a scoping review, the primary aim of this review is to examine the breadth of the body of literature on community-based interventions to address malnutrition (rather than estimates of their impact). As such, quality or similar assessment of studies, where the methodological rigor or risk of bias is assessed, is not a required part of such reviews (Peters, 2020; Munn et al., 2018). However, as part of the current review, we decided to additionally undertake a risk of bias assessment specifically for studies reporting quantitative results as a preliminary way of contextualizing the reported measures of impact on the outcomes reported. We decided not to perform similar quality assessments for qualitative studies as these studies (1) usually do not report the actual measure of impact (but rather investigate how interventions might be working), and (2) unlike quantitative measures of impact, there is not a single standard or a widely used method for quality assessment for qualitative studies. As such, given that any quality assessment is over and above the usual remit of scoping reviews, we do not feel that this introduces any limitations to the planned approach. We have added a brief statement clarifying this in the manuscript.

Method, lines 273 – 276:

“As scoping reviews are exploratory in nature, risk of bias assessments are not typically required as part of the guidelines for scoping reviews (Munn et al., 2018). However, we plan to assess the risk of bias among studies with an available quantitative measure as a preliminary way of contextualizing the reported measures of impact on the outcomes reported.”

Munn Z, Peters MDJ, Stern C, et al. Systematic review or scoping review? Guidance for authors when choosing between a systematic or scoping review approach. *BMC Med Res Methodol.* 2018 19;18:143. doi: 10.1186/s12874-018-0611-x.

Peters MDJ, Godfrey C, McInerney P, et al. Chapter 11: Scoping Reviews (2020 version). Aromataris E, Munn Z, editors. *JBIM Manual for Evidence Synthesis.* JBI; 2020. Available from <https://synthesismanual.jbi.global>. <https://doi.org/10.46658/JBIMES-20-12>.

3. Introduction

- Page 8, line 134: You say, “several systematic reviews” and cite only one reference that is not a systematic review. Please, adjust and include the original references.

Response: Thank you for pointing out this error. We have provided the following citations in the revised manuscript.

Salam RA, Hooda M, Das JK, et al. Interventions to Improve Adolescent Nutrition: A Systematic Review and Meta-Analysis. *J Adolesc Health.* 2016;59:S29-S39. doi: 10.1016/j.jadohealth.2016.06.022.

Lassi ZS, Moin A, Das JK, et al. Systematic review on evidence-based adolescent nutrition interventions. *Ann N Y Acad Sci.* 2017;1393:34-50 PubMed . doi: 10.1111/nyas.13335.

- Page 8, line 138: Include a reference for the following sentence “There is limited evidence that protein-energy supplements are effective for adolescents”.

Response: We have added the following reference to the sentence:

Lassi ZS, Moin A, Das JK, et al. Systematic review on evidence-based adolescent nutrition interventions. *Ann N Y Acad Sci.* 2017;1393:34-50 PubMed . doi: 10.1111/nyas.13335.

4. Methods

- Do you intend to conduct a scoping review or a systematic review? There are important differences among them, and your methods is not in accordance with a scoping review. Please, see the guidance of Joanna Briggs Institute for scoping reviews. Some points: PICO is not appropriate for scoping reviews; the recommended reporting guideline is PRISMA ScR instead of SWiN (as cited in the abstract and Methods).

Response: Thank you for this comment. We are conducting a scoping review, following the Joanna Briggs Institute (JBI) guidance and sample protocol for scoping reviews with some alterations. The use of PICO (rather than PCC) was one of these alterations. We acknowledge that this may have caused confusion and have therefore now amended this and checked the other relevant sections of the manuscript to ensure that we have covered all sections of the JBI guidance and template protocol (including the use of PCC).

Introduction, lines 163 - 166:

“We decided to conduct a scoping review as our primary aim was to summarize the overview of the evidence on community-based interventions for adolescents in LMICs, rather than to pursue a specific clinical or epidemiological question related to these or provide evidence to directly inform policy or practice (Munn et al., 2018).”

Methods, line 187 and Table 1:

“We will use the Participants, Concept, and Context (PCC) model (Table 1) to guide our search strategy.”

To note, we recognize that the SWIM approach is intended as an addition to the PRISMA checklist specifically for systematic reviews. In our original draft, we intended to state that we aimed to follow the SWIM guidelines as a method of meaningfully summarizing the results of the scoping review, rather than as an overall approach. Indeed, we have noted in at multiple points in this draft that we would follow the PRISMA-ScR reporting guidelines:

Methods, lines 244 – 246:

“A study flow diagram stating the specific reasons for exclusion will be maintained following the PRISMA for Scoping Review statement (PRISMA-ScR).”

Methods, lines 300 – 301:

“We will follow the PRISMA Extension for Scoping Reviews (PRISMA-ScR) checklist and guidelines to ensure a robust and replicable process.”

Methods, lines 304 – 305:

"The final protocol was registered prospectively with the Open Science Framework (<https://osf.io/t2d78>) on July 19, 2023, based on the PRISMA Extension for Scoping Reviews (PRISMA-ScR). "

- How did you define the eligible interventions? Will be the study included if it combines more than one intervention?

Response: Our scoping review will include single-domain or multifaceted nutrition and physical activity interventions conducted in community-based settings. In case a study includes interventions conducted in both school- and community-based settings, only the information related to community-based intervention will be included in the synthesis section.

- Why will you use Endnote and Covidence for the same purpose?

Response: We have adjusted the sentences, and we will use only the Covidence, as all the study members and reviewers have access to the Covidence.

Data management, lines 235 – 237:

"The records will be imported into Covidence (Veritas Health Innovation, Melbourne, Australia), an Internet-based systematic review management program. Detection and removal of duplicates, title and abstract screening, and full-text screening will be performed by Covidence."

- Where the data extraction will be performed?

Response: The extraction of data will be performed using an extraction form in Microsoft Excel (supplementary file 2)

- In the OSF is described as end date for this scoping review December 2023. Did you have already finished the study?

Response: The protocol was registered on OSF in July 2023 with an anticipated study completion time of 6 months. Therefore, the end date was initially set as December 2023. However, we would like to clarify that this review is still in progress.

- Please, include the date when the PubMed search was conducted, and the results obtained.

Response: The precise search strategies for all databases including any filters, dates, and limits used have been included in the supplementary file 1.

- Did you plan an update of your search strategy?

Response: There are no intentions to update the search strategy, as it has already been reviewed and approved by research librarians and experts in the domain.

REVIEWER	Mayasari, Noor Rohmah State University of Surabaya, Nutrition
REVIEW RETURNED	10-Mar-2024
GENERAL COMMENTS	All questions have been addressed by authors.